# A specific role for serotonin in overcoming effort cost

Florent Meyniel[1,2,3]*, Guy M Goodwin[4,5], JF William Deakin[6], Corinna Klinge[4,5], Christine MacFadyen[4,5], Holly Milligan[6], Emma Mullings[6], Mathias Pessiglione[1,2†], Raphaël Gaillard[7,8,9,10†]

[1]Motivation, Brain and Behavior team, Institut du Cerveau et de la Moelle épinière, Groupe Hospitalier Pitié-Salpêtrière, Paris, France; [2]INSERM UMRS 1127, CNRS UMR 7225, Université Pierre et Marie Curie (UPMC-P6), Paris, France; [3]Cognitive Neuroimaging Unit, NeuroSpin Center, INSERM U992, Institut d'Imagerie Biomédicale, Direction de la recherche fondamentale, Commissariat à l'énergie atomique et aux énergies alternatives, Gif-sur-Yvette, France; [4]Department of Psychiatry, University of Oxford, Oxford, United Kingdom; [5]Warneford Hospital, University of Oxford, Oxford, United Kingdom; [6]Neuroscience and Psychiatry Unit, The University of Manchester, Manchester, United Kingdom; [7]Centre Hospitalier Sainte-Anne, Service de Psychiatrie, Paris, France; [8]Faculté de Médecine Paris Descartes, Université Paris Descartes, Sorbonne Paris Cité, Paris, France; [9]Laboratoire de Physiopathologie des maladies Psychiatriques, Centre de Psychiatrie et Neurosciences, INSERM U894, Université Paris Descartes, Sorbonne Paris Cité, Paris, France; [10]Human Histopathology and Animal Models, Department of Infection and Epidemiology, Institut Pasteur, Paris, France

*For correspondence: florent.
meyniel@gmail.com

†These authors contributed
equally to this work

Competing interest: See
page 15

Reviewing editor: Joshua I
Gold, University of Pennsylvania,
United States

**Abstract** Serotonin is implicated in many aspects of behavioral regulation. Theoretical attempts to unify the multiple roles assigned to serotonin proposed that it regulates the impact of costs, such as delay or punishment, on action selection. Here, we show that serotonin also regulates other types of action costs such as effort. We compared behavioral performance in 58 healthy humans treated during 8 weeks with either placebo or the selective serotonin reuptake inhibitor escitalopram. The task involved trading handgrip force production against monetary benefits. Participants in the escitalopram group produced more effort and thereby achieved a higher payoff. Crucially, our computational analysis showed that this effect was underpinned by a specific reduction of effort cost, and not by any change in the weight of monetary incentives. This specific computational effect sheds new light on the physiological role of serotonin in behavioral regulation and on the clinical effect of drugs for depression.
Clinical trial Registration: ISRCTN75872983

## Introduction

Selective serotonin reuptake inhibitors (SSRI) are the most prescribed medications for major depressive episodes (*Bauer et al., 2008*). The effects of SSRI on improving mood and reducing anxiety have been well documented (*Trivedi et al., 2006*; *Stahl, 2008*; *Cipriani et al., 2009*) and many experimental studies showed that serotonin modulates emotional processing even in healthy volunteers (*Harmer et al., 2009*; *Serretti et al., 2010*). Loss of interest or pleasure in daily activities is also a key symptom of depression (*American Psychiatric Association, 2013*). However, the impact

**eLife digest** Neuromodulators are chemicals released in the brain that affect the activity of brain cells. Serotonin is a neuromodulator with the most complicated role: it is released in most brain regions and affects behavior in diverse ways. Serotonin is implicated in the regulation of mood, anxiety, impulsivity and learning. Moreover, most medications for depression target serotonin. A lack of motivation is an important symptom of depression, but exactly how serotonin affects motivation still remains unclear.

Meyniel et al. studied how increasing the amount of serotonin in the brain affects motivation in healthy people. The volunteers in the experiments squeezed a handgrip: the longer they squeezed, the more money they got as a reward. Before the experiment, some of the volunteers received an antidepressant drug that increases the amount of serotonin surrounding their brain cells, while others received a placebo.

The experiments revealed that, compared to the people who had the placebo, those who received the drug put in more effort to get a reward. More serotonin could increase motivation by reducing the perceived cost of putting in more effort, or by making people value the reward more. A mathematical model of the results showed that the increased motivation in the antidepressant group was more consistent with serotonin reducing the cost of putting in an effort, rather than increasing how much the reward was valued.

Combined with previous findings, these results suggest that serotonin affects the processing of cost associated with tasks – be that the amount of effort required, delays in getting a reward, or a punishment. Further experiments are now required to understand if the same mechanism operates in people with depression, and if so, whether it can be altered to promote recovery. It will also be important to better understand the interaction between serotonin and other neuromodulators such as dopamine.

of SSRI on the motivation deficit (apathy) remains rather controversial (*Papakostas et al., 2006*; *Weber et al., 2009*).

Indeed, several studies reported that SSRI treatments do not reduce apathy as much as other symptoms (*Fava et al., 2014*) or can even *induce* apathy in patients (*Barnhart et al., 2004*; *Sansone and Sansone, 2010*; *Padala et al., 2012*). However, other studies also reported that SSRI treatments increase motivation or at least the sensitivity to reward (*Tang et al., 2009*; *Stoy et al., 2012*; *Yuen et al., 2014*). The related animal literature is also contradictory: some studies on cost/benefit trade-off showed reduced effort expenditure with SSRIs (*Yohn et al., 2016*) or no effect with serotonin blockers (*Denk et al., 2005*), while other studies reported increased motor activity induced by SSRI (*Weber et al., 2009*) or optogenetic stimulation of the dorsal raphe nucleus (*Warden et al., 2012*).

One issue with clinical studies is that SSRI effects may be confounded by an interaction with the pathological state: an SSRI treatment was reported to decrease apathy in late-life depression (*Yuen et al., 2014*) and to increase it in Parkinson's disease (*Zahodne et al., 2012*). Another issue is that apathy may not be a simple construct. It is typically captured as a loss of behavioral activation (*Cléry-Melin et al., 2011*; *Treadway et al., 2012*), which could arise from different causes at the cognitive level: for instance a diminished sensitivity to reward attractiveness, or alternatively an exacerbated sensitivity to action cost. The ambition of the present study therefore was two-fold: we aimed (1) to clarify the effects of an SSRI on the motivation of effortful behavior in healthy volunteers, unconfounded by psychopathology, and (2) to reconcile these effects with a more general role of serotonin in the sensitivity to action costs vs. benefits.

Although experimental findings seem diverse at first sight, they might suggest a general role for serotonin in the behavioral adaptation to action costs. Previous studies suggested that serotonin is implicated in the motor aspects of action production and also the valuation processes that motivate the action. Notably, serotonin was found to condition impulse control and the capacity for behavioral inhibition (*Cools et al., 2005*; *Crockett et al., 2009*; *Warden et al., 2012*; *Guitart-Masip et al., 2014*) and to promote patience for delayed rewards (*Schweighofer et al., 2008*;

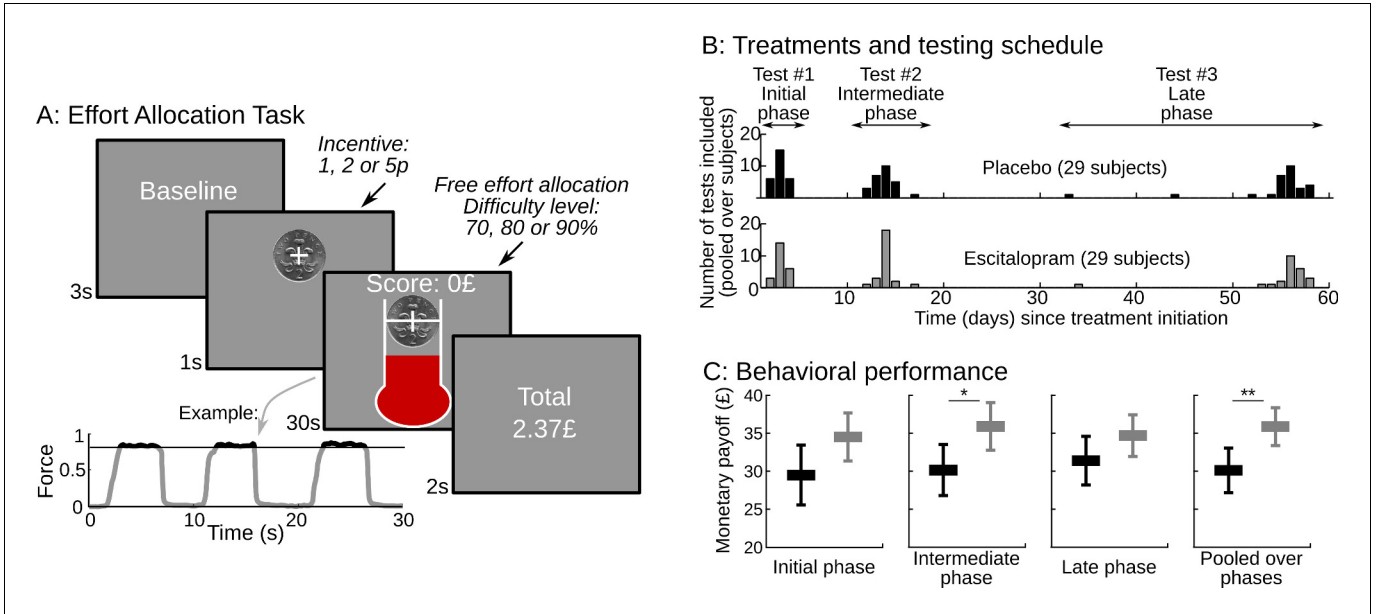

**Figure 1.** Task design and behavioral performance. (**A**) The screenshots depict a trial as it was presented to subjects. Subjects were free to allocate their effort as they wished over the 30s corresponding to the trial duration. They were instructed that their monetary payoff would be proportional to both the monetary incentive and the effort duration, i.e. the time spent squeezing a handgrip harder than a target force level, which varied with task difficulty. Subjects were provided with on-line feedback on the payoff accumulated in the trial (score on the top) and on the instantaneous pressure exerted on the grip (fluid level in the thermometer). The force time series of an example trial is shown below the screenshots, revealing 3 effort periods, with rewarded effort (force above target) plotted in black (not gray). Two factors were manipulated across trials: (i) the incentive level, shown as a coin image (1, 2 or 5p) and (ii) the difficulty level, corresponding to the same white bar in the thermometer reached with different target force levels (70%, 80% or 90% of the maximal force). The last screen summarized the payoff cumulated over preceding trials. (**B**) Using a double-blind procedure, healthy subjects were assigned to one of the two treatment groups, corresponding to a daily intake of either placebo or escitalopram (10 mg during the initial phase, 20 mg during the intermediate and late phase) during 9 weeks. Each subject completed the effort allocation task three times at distinct treatment phases (initial, intermediate and late). Numbers of subjects and visits correspond to data sets included in the analysis after compliance and quality checks. (**C**) The three left-most graphs show task performance (as reflected in monetary payoff) sorted by treatment group (black: placebo; gray: escitalopram) and time since treatment onset. Statistical significance was assessed with two-sample, two-sided t-tests. On the right-most plot, payoff was averaged over visits at the subject level. Statistical significance was assessed with ANOVAs including treatments as between-subject factors and test phase (initial, intermediate or late) as a within-subject factor. *p<0.05; **p<0.005. Error bars indicate Student's 95% confidence intervals.

The following source data is available for figure 1:

**Source data 1.** The MATLAB data file contains the payoff earned by each participant at each visit, in the placebo and escitalopram groups.

*Miyazaki et al., 2014*; *Worbe et al., 2014*). Serotonin also impacts the determinants of actions, for instance, how positive and negative outcomes guide learning in humans (*Chamberlain et al., 2006*; *Crockett et al., 2009*; *Faulkner and Deakin, 2014*), monkeys (*Clarke et al., 2004*) and rodents (*Bari et al., 2010*). Several attempts have been made previously to capture serotonin function in a coherent computational theory. Key ideas are that serotonin (1) regulates the impact of action costs such as punishments (*Daw et al., 2002*; *Niv et al., 2007*) or delay (*Niv et al., 2007*; *Cools et al., 2011*; *Miyazaki et al., 2014*) and (2) adjusts the propensity to activate versus inhibit the action (*Boureau and Dayan, 2011*; *Cools et al., 2011*).

Borrowing from these models, we retain the general working hypothesis that serotonin regulates behavioral activation by modulating the weight of action costs rather than benefits.

We further suggest that the notion of action cost could be more general than initially envisaged. Indeed, in the literature quoted above, actions are mostly implemented as binary responses (e.g., approach vs. avoidance) and their costs or benefits are manipulated through the valence of their outcome (e.g., monetary gain or loss). Such tasks therefore over-simplify the real-life situation, where actions may often require more or less effort, depending on their intensity and duration.

Here, we tested the specific hypothesis that serotonin regulates the weight of effort cost in action production, as opposed to the weight of expected benefit. To assess the impact of changing cerebral serotonin levels and how it unfolds over time, we compared different groups of healthy subjects treated under double blind conditions with escitalopram (an SSRI) or placebo during eight weeks. Participants performed a previously published task (*Meyniel et al., 2013*), during which they allocated physical effort over time in order to maximize a monetary payoff that increased with effort duration (see *Figure 1A*). The task thus involves trading a physical effort cost against distinct levels of monetary incentives. Although changes in cost and benefit were independently manipulated in the task, they may result in intricate effects at the behavioral level. To disentangle between potential effects of serotonin on cost or benefit estimates, or both, we used a formal model of how decisions are generated by a hidden level of computations in our task, as previously described (*Meyniel et al., 2013, 2014*). Based on this computational analysis of the behavior, we could pinpoint the specific effect of serotonin on cost-related parameters, and track the predicted effects of such a computational change onto the experimental measures.

## Results

### Escitalopram improves performance in the effort allocation task

58 subjects were included in the analysis, see *Table 1*. We took the cumulated payoff as the primary measure of performance in the task and compared it between groups and visits (see *Figure 1C*). We found a significant effect of treatment group: performance was significantly improved in the escitalopram group ($F_{1, 58.4}=9.37$, $p=0.003$) as compared to the placebo group. This difference was stable over time: there was no significant interaction between groups and visits ($F_{2, 91}=0.22$, $p=0.8$). The average payoff per visit was £35.9±1.21 s.e.m. and £30.1±1.44 s.e.m. in the escitalopram and placebo groups respectively.

### Escitalopram has a specific computational effect on effort cost

The better performance observed in the escitalopram group could be underpinned by different mechanisms: alleviation of effort costs or inflation of incentive values, or both. To disentangle between the two mechanisms, we relied on a computational model of effort allocation that was previously proposed (*Meyniel et al., 2013*; *Meyniel et al., 2014*). This model assumes that a single computational variable, termed cost evidence, accounts for the decision to stop and resume effort exertion. Cost evidence waxes during effort (with slope Se) until reaching an upper bound where effort is stopped, and it wanes during rest (with slope Sr) until reaching a lower bound where effort is resumed. The distance between bounds is the cost-evidence amplitude (denoted A). Effort and rest durations are determined by the ratios of amplitude and slopes (see *Figure 2A*) so that performance depends on the value of latent parameters (A, Se, Sr) and its potential modulation by task factors (monetary incentive and effort difficulty). We used a Bayesian Model Selection procedure previously validated (*Meyniel et al., 2013, 2014*) to pinpoint the effect of the task factors onto these latent parameters (see Materials and methods).

Replicating our previous studies, the best model showed that incentives impacted the amplitude (A) and dissipation slope (Sr), whereas effort difficulty impacted the accumulation slope (Se), as illustrated in *Figure 2A*, right. To capture all the effects, the cost-evidence accumulation model therefore necessitates five free parameters: the mean slope of cost-evidence accumulation ($Se_m$) and its steepening for higher difficulty levels ($Se_d$); the mean slope of cost-evidence dissipation during rest ($Sr_m$) and its steepening for higher incentives ($Sr_i$); and the expansion of the cost-evidence amplitude as higher incentives push the bounds back ($A_i$). The fact that the same best model was found independently in each treatment group with high confidence levels (exceedance probabilities xp>98% in each group) indicates that all subjects can be characterized within this common computational framework. We computed for each subject the best-fitting values of the model parameters ($A_i$, $Se_m$, $Se_d$, $Sr_m$, $Sr_i$). Since the interaction between treatments (placebo vs. escitalopram) and treatment phase was not significant for any parameter (all $F_{2, 91}<2.03$, $p>0.14$), we provide fitted values pooled over treatment phases in *Figure 2B*.

Only one model parameter was significantly different between the placebo and escitalopram groups: the cost-evidence accumulation slope ($Se_m$, $F_{1, 58.0}=9.88$, $p=0.003$). This parameter captures

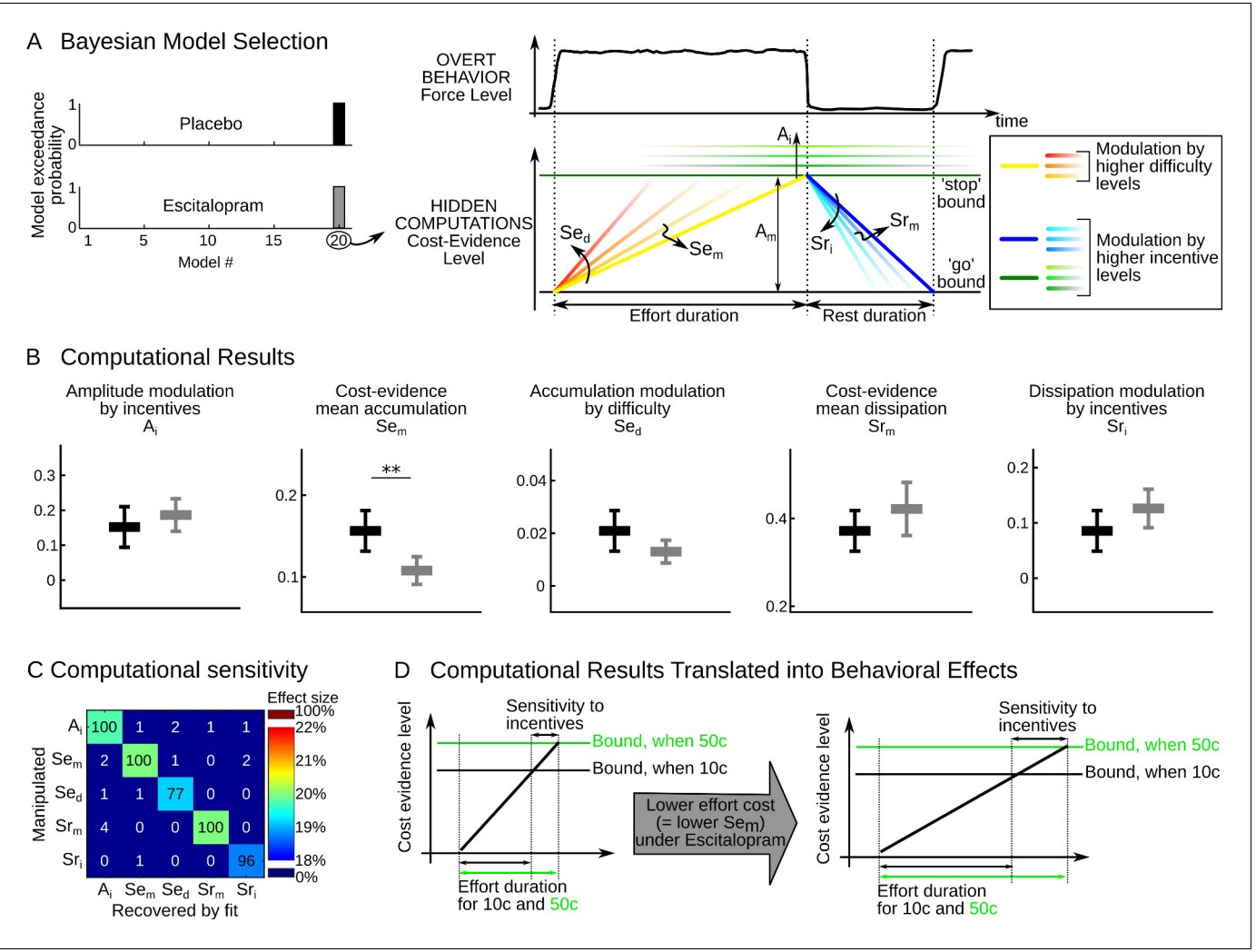

**Figure 2.** Computational results. (**A**) The cost-evidence accumulation model assumes that effort and rest durations are respectively determined by the accumulation (mean slope $Se_m$) and dissipation (mean slope $Sr_m$) of cost evidence between bounds (mean amplitude $A_m$). Possible modulations of these parameters by incentive and difficulty levels were implemented in 20 distinct models. In the best model identified (#20) by Bayesian selection, increasing effort difficulty shortens effort duration by steepening the accumulation slope (a parametric effect controlled by parameter $Se_d$ and illustrated with colors from yellow to red). Increasing the incentive level has two effects: first, it shortens rest duration by speeding up the dissipation (parametric effect of $Sr_i$, illustrated by colors from dark to light blue); second, it lengthens effort duration by pushing back the bounds (parametric effect of $A_i$, illustrated by green scaling). (**B**) Plots show inter-subject means and Student's 95% confidence intervals obtained for the fitted values of model parameters (which were averaged over visits at the subject level). To facilitate visual comparison, scales and offsets were adjusted so that mean and error bars are visually equal across plots in the placebo group. Statistical significance corresponds to ANOVAs including treatment group (escitalopram vs. placebo) as a between-subject factor and treatment phase as a within-subject factor (initial, intermediate or late); **p<0.005. (**C**) Data in the placebo group served as a baseline to simulate effort and rest durations after imposing a 20% increase in computational parameters. In the table, each row corresponds to a simulated change in one single parameter. Colors denote the effect sizes recovered by model fitting for each parameter, as percent of change compared to baseline. Numbers indicate the percentage of 'hit' (on the diagonal) and 'false alarm' (off-diagonal) in detecting a significant change in parameter values with a paired t-test thresholded at p<0.01. (**D**) The graph illustrates why the effect of escitalopram, characterized at the computational level as a reduced accumulation slope of cost-evidence during effort ($Se_m$), should translate at the behavioral level into both a longer effort duration and an increased sensitivity of effort duration to incentive level.

The following source data is available for figure 2:

**Source data 1.** The MATLAB data file contains the fitted value of parameters $A_i$, $Se_m$, $Se_d$, $Sr_m$, $Sr_i$ (see Materials and methods, *Equation 2*), for each participant at each visit, in the placebo and escitalopram groups.

**Table 1.** Details on participants N corresponds to the number of subjects per treatment type and phase. A few datasets were not available due technical problems and late withdrawals. Based on criteria specific to the present task (and not to the clinical trial), some subjects were excluded from the analysis ('excluded'). We report the age and sex of participants included in the analysis and the exact time of their test since the treatment onset.

| Treatment type | Treatment phase | N not available | N excluded | N after exclusion | Sex (Male / female) | Age (years) ± SD of included subjects | Time since treatment onset (days) ± SD for included subjects |
|---|---|---|---|---|---|---|---|
| Placebo | Initial | 0 | 5 | 27 | 14/13 | 23.4 ± 4.35 | 3.0 ± 0.68 |
| Placebo | Intermediate | 0 | 6 | 26 | 14/12 | 23.2 ± 4.33 | 13.8 ± 1.13 |
| Placebo | Late | 0 | 4 | 28 | 15/13 | 23.4 ± 4.27 | 54.7 ± 4.98 |
| Escitalopram | Initial | 1 | 8 | 23 | 11/12 | 24.5 ± 4.71 | 3.1 ± 0.63 |
| Escitalopram | Intermediate | 1 | 6 | 25 | 12/13 | 24.5 ± 4.51 | 14.0 ± 0.87 |
| Escitalopram | Late | 2 | 6 | 24 | 10/14 | 24.6 ± 4.61 | 55.3 ± 4.69 |

the average value of effort cost across conditions: there was a 31% decrease in the escitalopram group compared to the placebo group (on average with ± s.e.m., placebo: 0.16±0.012, escitalopram: 0.11±0.008); the difference was actually significant at each visit (all p<0.015; two-sided t-test).

The difference between escitalopram and placebo never reached significance for the other model parameters (all p>0.08, see *Table 2*; on average with ± s.e.m. for placebo vs. escitalopram, $A_i$: 0.15±0.028 vs. 0.19±0.022, $Se_d$: 0.021±0.004 vs. 0.013±0.002, $Sr_m$: 0.37±0.022 vs. 0.42±0.029, $Sr_i$: 0.086±0.018 vs. 0.126±0.016). We checked the sensitivity and specificity of our model fitting procedure in detecting a treatment effect through simulations. A 20% change in any given parameter was reliably detected and the difference recovered by model fitting was significant in a 96% of simulations at least for $A_i$, $Se_m$, $Sr_m$ and $Sr_i$ and in a 77% of simulations for $Se_d$. Importantly, a change in one single parameter was recovered without propagating to other parameters and the false alarm rate was below 5% for all parameters (*Figure 2C*).

To further test the specificity of SSRI effect on cost-evidence accumulation, we performed another Bayesian model selection that contrasted the two groups of subjects. Parameters of the cost-evidence accumulation model were fitted on the placebo group data to serve as a reference. In the

**Table 2.** Treatment effect on computational parameters and behavioral measures.
All numbers are p-values obtained from ANOVAs. p-values lower than 0.05/5 = 0.01 (computational parameters) and 0.05/6 = 0.008 (behavioral measures) appears in bold to show significant effects that survive correction for multiple comparisons.

| Variable | Treatment | Visit | Interaction |
|---|---|---|---|
| $A_i$ | 0.388 | 0.197 | 0.406 |
| $Se_m$ | **0.003** | 0.039 | 0.260 |
| $Se_d$ | 0.0797 | 0.543 | 0.137 |
| $Sr_m$ | 0.186 | 0.778 | 0.612 |
| $Sr_i$ | 0.130 | 0.196 | 0.557 |
| Effort duration – mean | **0.002** | 0.199 | 0.875 |
| Effort duration – sensitivity to incentive | 0.023 | 0.187 | 0.115 |
| Effort duration – sensitivity to difficulty | 0.247 | 0.813 | 0.318 |
| Rest duration – mean | 0.213 | 0.482 | 0.531 |
| Rest duration – sensitivity to incentive | 0.162 | 0.937 | 0.807 |
| Rest duration – sensitivity to difficulty | 0.115 | 0.423 | 0.681 |

**Table 3.** Model comparison assessing the specificity of treatment effect.

Data in the escitalopram group were fitted with the cost-evidence accumulation model. The parameters were fixed to the values fitted onto the placebo group, excepted when a modulation was permitted. The first row contains models that permit the modulation of one single parameter, whereas the remaining rows correspond to models that permit a combination of two modulations. Each cell gives log Bayes Factor (i.e. log model evidence) relative to the null model. Higher values denote better models.

| | $Se_m$ | $Se_d$ | $Sr_m$ | $Sr_i$ | $A_i$ |
|---|---|---|---|---|---|
| Only one modulation | 173.6 | 19.9 | −4.2 | −3.8 | 3.7 |
| Also includes $Se_m$ | | 176.6 | 169.6 | 169.7 | 170.7 |
| Also includes $Se_d$ | | | 15.7 | 16.1 | 21.7 |
| Also includes $Sr_m$ | | | | −7.3 | −1.1 |
| Also includes $Sr_i$ | | | | | 1.2 |

escitalopram group, they were fixed to these reference values and only one or two modulations were permitted to capture the treatment effects. Models including a modulation of $Se_m$ outperformed the others (log Bayes Factor, all $\Delta > 147.9$), see *Table 3*. A version with modulations of both $Se_m$ and $Se_d$ was slightly more likely than a version with only a modulation of $Se_m$ ($\Delta = 3$) and much more likely than any other combination ($\Delta > 5.9$).

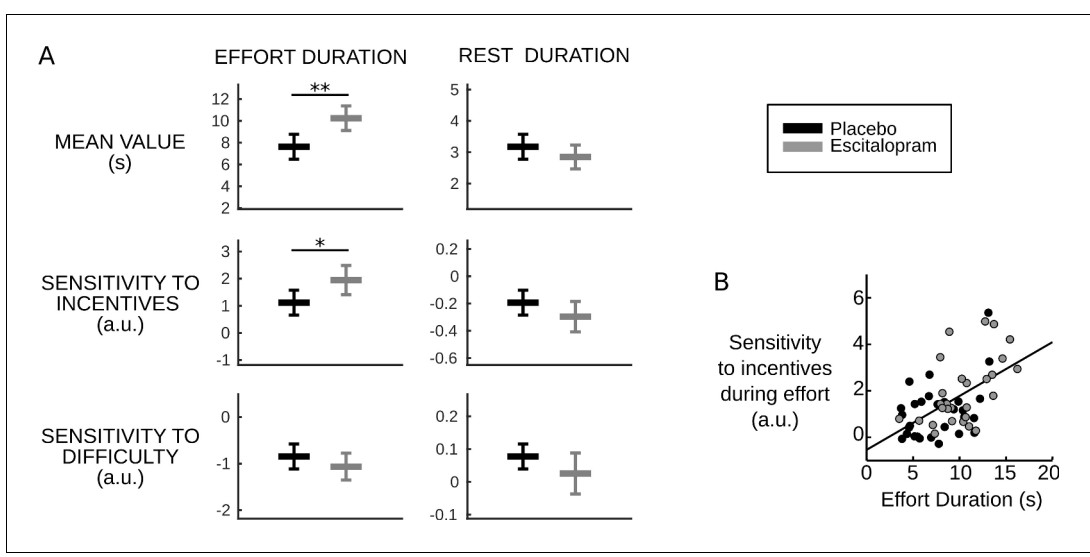

**Figure 3.** Behavioral results. (**A**) Plots show inter-subject means and Student's 95% confidence intervals obtained from linear regression.Regression coefficients were averaged over visits at the subject level. To facilitate visual comparison, scales and offsets were adjusted so that mean and s.e.m. are visually equal across plots in the placebo group. Statistical significance corresponds to ANOVAs including treatment group (escitalopram vs. placebo) as a between-subject factor and treatment phase as a within-subject factor (initial, intermediate or late); *p<0.05, **p<0.005. (**B**) As predicted by the cost-evidence accumulation model, effort duration and its sensitivity to incentive level are correlated across subjects (one dot corresponds to one subject; values were averaged across visits for each subject). The line shows the linear regression fit obtained when pooling the two treatment groups ($\rho_{56} = 0.55$, $p < 10^{-5}$).

The following source data is available for figure 3:

**Source data 1.** The MATLAB data file contains a description of the behavior obtained by linear regressions for each participant at each visit, in the placebo and escitalopram groups.

Thus, the specific effect of SSRI on cost-related parameters supports our hypothesis that serotonin is involved in the estimation of action cost, rather than benefits.

## Computational effects of escitalopram translate into behavioral effects

Reducing the effort cost ($Se_m$) only should have two effects at the behavioral level (see *Figure 2D*). The first is straightforward: the duration of effort epochs should be longer. The second is less trivial: the sensitivity of effort duration to incentive level should be increased. This is because in our model, incentive level modulates the amplitude between bounds (as captured by parameter $A_i$). Thus, the effect of incentive level on effort duration is proportional to the accumulation slope $Se_m$: if the accumulation is slower, then a given displacement of the upper bound will have a larger effect on effort duration.

Fulfilling these predictions, we found both a longer effort duration per se ($F_{1, 58.1}$=10.72, p=0.002, on average placebo: 7.63s±0.56, escitalopram: 10.2s±0.55, with s.e.m.) and a higher sensitivity of effort duration to incentive level ($F_{1, 58.5}$=5.46, p=0.023, on average with ± s.e.m., placebo: 1.11±0.22, escitalopram: 1.95±0.26) in the escitalopram group compared to placebo (see *Figure 3A*). The difference between the two treatment groups never reached significance for the other behavioral variables (all p>0.11, on average with ± s.e.m., for placebo vs. escitalopram, effort sensitivity to difficulty: −0.84±0.13 vs. −1.06±0.14, rest duration: 3.17s±0.19 vs 2.85s±0.19, rest sensitivity to incentives: −0.19±0.04 vs. −0.30±0.05, rest sensitivity to difficulty: 0.077±0.018 vs. 0.025±0.03).

Crucially, because they arise from a common cause, the model also predicts that the two effects should be correlated: the more $Se_m$ is reduced, the more effort duration and its sensitivity to incentive level should be increased. The Pearson correlation over subjects was indeed significantly positive in both groups (placebo: $\rho_{27}$=0.43, p=0.02; escitalopram: $\rho_{27}$=0.54, p=0.002, see *Figure 3B*).

## Discussion

Our behavioral results in healthy volunteers show that the SSRI escitalopram improves global performance and hence payoff in a task that involves trading effort cost against monetary benefit. Taking advantage of the independent manipulation of cost and benefit, our computational analysis characterized the effect of escitalopram as a specific diminution of effort cost.

Together with previous findings, our results support the hypothesis that cost may be a general functional domain for serotonin (*Boureau and Dayan, 2011*; *Dayan, 2012*). For instance, interventions targeting serotonergic transmission during probabilistic and reversal learning paradigms in rodents, monkeys and humans suggested that serotonin impacts sensitivity to negative rather than positive feedback (*Clarke et al., 2004*; *Chamberlain et al., 2006*; *Crockett et al., 2009*; *Bari et al., 2010*; *Cohen et al., 2015*). The reduction of serotonergic transmission in humans following acute tryptophan depletion (for a review, see *Faulkner and Deakin, 2014*) reduced information sampling in a decision-making task, but crucially, only when sampling had a financial cost (*Crockett et al., 2012*). Serotonin may also impact moral costs, such as unfairness in social decision-making (*Crockett et al., 2008*). Furthermore, serotonin may control the cost of delay in reward delivery in rodents, such that a higher firing rate of serotonergic neurons correlates with an increased ability to wait for bigger rewards (*Miyazaki et al., 2014*), without affecting the sensitivity to the reward itself (*Fonseca et al., 2015*). Conversely, low serotonin levels, e.g. after acute tryptophan depletion, were suggested to exacerbate sensitivity to the cost of waiting, which could result in impulse control disorders (*Crockett et al., 2009*; *Cools et al., 2011*; *Dayan, 2012*). Thus, it is tempting to build a parsimonious computational theory on the idea that serotonin is involved in processing all kinds of action costs. This is compatible with the notion that different types of costs are processed by distinct neural circuits, since neuromodulators such as serotonin can affect many brain regions. However, such a generalization would necessitate assessing the putative role of each mono-amine and their interactions with different types of costs within the same study. Previous attempts in rodents have observed effects that are inconsistent with the present results: in cost/benefit trade-off tasks, serotonin blockers impacted choices only when costs were delays, and not physical efforts (*Denk et al., 2005*), whereas dopamine blockers affected both types of costs. Such discrepancies may call for caution when translating the results of pharmacological studies in rodents into medication effects in humans.

A general role for serotonin in cost processing also seems compatible with the effects on apathy and impulsivity reported in healthy and pathological conditions (*Cools et al., 2005*; *Papakostas et al., 2006*; *Schweighofer et al., 2008*; *Crockett et al., 2009*; *Weber et al., 2009*; *Warden et al., 2012*; *Guitart-Masip et al., 2014*; *Miyazaki et al., 2014*; *Worbe et al., 2014*). Indeed, according to our interpretation, SSRIs might reduce the perceived cost of performing actions, which would promote behavioral activation, hence the alleviation of apathy. As SSRIs might also reduce the perceived cost of delaying action, they would improve response control and reduce impulsivity (*Cools et al., 2005*; *Cools et al., 2011*; *Crockett et al., 2009*; *Dayan, 2012*; *Bari and Robbins, 2013*; *Miyazaki et al., 2014*; *Fonseca et al., 2015*).

However, it is important to note that the SSRI effects obtained here relate to effort cost and not to time discounting. Although time is central in our task, it departs from delay and waiting paradigms in several respects. First, the overall task duration was fixed and its pace was independent from how subjects allocated their effort within trials. Second, the reward was gained instantaneously, concomitantly to effort production, without delay. Therefore, the SSRI effect on effort duration (how long subjects sustain an effort) cannot be explained by a change in temporal discounting (how long subjects wait for a reward).

The direction of the effect obtained in the present study suggests that higher serotonin level alleviates the effort cost. This interpretation is based on the assumption that the primary effect of SSRI is to increase serotonin level in the synaptic cleft (*Nutt et al., 1999*; *Stahl, 2008*). However, the net effect of SSRI might not be so straightforward to interpret at the molecular level for several reasons. First, a high tonic concentration could have the paradoxical effect of reducing the sensitivity to phasic serotonergic signals (*Faulkner and Deakin, 2014*). Second, the ubiquitous negative feedback regulation by auto-receptors can initially revert the response expected from high extracellular levels of serotonin (*Stahl, 2008*; *Fischer et al., 2014*) and also produce non-linear dose-response functions for cognitive performance (*Bari et al., 2010*). Third, much evidence suggests that different serotonin projections to different forebrain systems mediate varying acute and chronic adaptive responses to aversive events (*Hale et al., 2013*; *Faulkner and Deakin, 2014*). The effect of serotonin also depends on the location in the brain and on the type of receptors, for instance, 5-HT1A and 5-HT2A receptors respectively inhibit and excite motoneurons (*Jacobs and Azmitia, 1992*). Thus, the superficial interpretation that serotonin helps overcoming effort costs will need to be refined by addressing the complexity of SSRI effects at the molecular level. We nevertheless note that the superficial interpretation is consistent with demonstrations that serotonin also helps overcoming other costs such as delay in humans (*Schweighofer et al., 2008*).

Refinement is also needed at the computational level. Our model does not specify how exactly serotonin could attenuate the impact of effort cost on action production. An indirect effect through an increase in muscular capacity can be excluded since there was no difference in maximal force between the placebo and SSRI groups at any visit. The computational analysis simply suggests that a slower accumulation of effort cost (lower $Se_m$) under SSRI prolonged effort duration. It does not distinguish between down-regulation of the cost signal itself, or down-regulation of the weight this cost has in the decision to produce an effort. The former (perceptual) view would be in line with an analgesic effect of escitalopram, which is consistent with the findings that nociception and/or somatosensory perception are both modulated by serotonin at central levels (*Jacobs and Azmitia, 1992*), that serotonin is targeted by common drugs modulating pain like acetaminophen (*Smith, 2009*) and that serotonin modulates the inhibitory feedback loop that allows muscular fatigue to down-regulate the motor command (*Gandevia, 2001*; *Cotel et al., 2013*). The latter (decisional) view would be more generalizable to the capacity of overcoming other types of costs.

As the serotonin and dopamine systems strongly interact (*Dremencov et al., 2009*; *Cools et al., 2011*; *Schilström et al., 2011*; *Fischer et al., 2014*), the SSRI effect observed here might also be at least in part mediated by dopamine. Indeed, the beneficial effect of SSRI shown here is reminiscent of previous reports about dopaminergic manipulations in mesolimbic structures, which affected the propensity to choose high reward – high effort over low reward – low effort options in rodents (*Cousins et al., 1996*; *Salamone et al., 2007*). However, if SSRIs antagonize dopamine release (*Dremencov et al., 2009*; *Cools et al., 2011*), escitalopram should have reduced effort production in our task (but see *Schilström et al., 2011*). Moreover, the fact that escitalopram did not modulate incentive effects on parameters such as the amplitude or dissipation slope argues against a participation of dopamine. Indeed dopaminergic manipulations in humans have been repeatedly shown to

affect reward processing, not only in learning contexts (*Frank et al., 2004*; *Pessiglione et al., 2006*; *Palminteri et al., 2009*) but also in reward/effort trade-off paradigms (*Wardle et al., 2011*; *Treadway et al., 2012*; *Le Bouc et al., 2016*). It might be that serotonin and dopamine have complementary roles in promoting action production (and alleviating apathy): the former by reducing effort cost and the latter by enhancing the incentive value of potential rewards.

Previous studies suggest that we should distinguish between acute and chronic effects of SSRIs (*Fischer et al., 2014*). We did not find an interaction with time in our results but only a main effect of treatment group. Both parametric and non-parametric statistics showed that such a group difference is very unlikely to arise from chance in sampling the population (under the null hypothesis). General psychological assessment at baseline was also similar in the two groups. However, we acknowledge that our shortest time since treatment onset (3 days) may already depart from the acute regime, that ceiling effects in the SSRI group could in principle mask an interaction with time, and that our study may lack the statistical power to reveal such an effect. This absence of an interaction contrasts with the apparently delayed clinical effect of SSRIs in depressed patients, which usually take weeks to improve mood (*Stahl, 2008*). However, at the molecular level, serotonin release is boosted by SSRIs from treatment onset (*Kobayashi et al., 2008*) and SSRI effects on emotional processing also occur with little or no delay in healthy subjects (*Harmer et al., 2003*). In pathological conditions, time may be needed to adjust to the new, less negative perception of costs, as well as the reduced emotional bias, and convert these implicit changes into a conscious subjective improvement that can be reported to the practitioner (*Harmer et al., 2009*; *Cools et al., 2011*). This idea has been formalized in a recent study showing how positive and negative outcomes can shape mood on the long run (*Eldar and Niv, 2015*). Further studies in depressed patients are needed to assess whether an early detection of effort cost attenuation could be used to predict long-term treatment effects on clinical symptoms.

Finally, our results also indicate that non-trivial behavioral effects can be accounted for by a change in a single computational parameter ($Se_m$): a specific modulation of cost can result in both longer efforts and an increased sensitivity of effort duration to potential benefits in our results. Intuitively, this effect on reward sensitivity corresponds to the idea that when perceived costs are too high, a change in reward prospect will have little effect. Thus, our analysis shows that the behavioral consequences of cost and benefit estimates are intricate but computationally tractable. Our paradigm and model could provide experimental and conceptual tools to refine the description of motivational disorders. Distinct dysfunctions, such as amplification of effort cost vs. flattening of reward prospects, might call for different treatments: if drugs modulating serotonin only affect cost estimates, then other drugs (possibly dopaminergic) should aim to impact the valuation of potential benefits.

## Materials and methods

### Participants

Healthy volunteers (18–45 years old) were recruited to the study by public advertisement after approval by the Ethics Committee of Berkshire, UK (protocol CL1-20098-81, on 28th Sept. 2011) and registration as a clinical trial, ISRCTN75872983. Participants gave their written informed consent prior to participation. Normal health was checked by clinical and psychiatric examinations including laboratory tests. Exclusions and withdrawals from the study were adjusted to include 64 participants (2 treatment groups of 32 participants, 16 men per group), tested at 3 separate visits, resulting in 192 completions of the task. Given that the clinical trial was exploratory and also included tests from other research groups, the sample size was not selected specifically for our study; however it appeared reasonable given typical studies in the field. Indeed, published between-subject comparisons of placebo and anti-depressant treatments were made on a lower sample size per group (e.g. N=14 in *Harmer et al., 2004*, N=20 in *Chamberlain et al., 2006* and N=30 in *Guitart-Masip et al., 2014*), and a single visit per subject (while we have three). Due to technical problems or late withdrawals, 4 task completions were not fully acquired and therefore unusable. The remaining data were checked for quality by F.M. prior to unblinding: 11 were excluded due to mis-calibration of task difficulty; 9 due to hardware default or signal quality; 15 for non-compliance with task instructions. Subjects were asked to produce an effort in every trial. As there were eight trials per condition

in the task, a given visit was excluded for non-compliance when the total number of effort (or rest) was lower than three in at least one condition. As a result, the number of participants per visit varied between 23 and 28 in each treatment group, for a total of 153 task completions. Note that a given participant may produce an interpretable dataset at a given visit and not at another. The total number of participants with at least one interpretable dataset across the three visits happened to be 29 in each treatment group. The finally included data set is summarized in *Table 1*.

Sex ratio (14/29 vs. 16/29, Z test for proportion: z= 0.53, p=0.6), ratio of excluded data sets (15/96 vs. 20/92, z=1.08, p=0.28) and age (mean with ± s.e.m., 23.2 ± 0.8 vs. 24.2 ± 0.8, p=0.38) were similar in the two groups. We also used psychological tests to assess differences between groups of subjects at baseline, before treatment. T-test comparison showed no significant difference in Mood Visual Analog Scales for the items 'happy' (p=0.75), 'sad' (p=0.96), 'hostile' (p=0.48), 'alert' (p=0.73), 'anxious' (p=0.51), 'calm' (p=0.14), nor in the Hospital Anxiety Depression score for anxiety (p=0.68) and depression (p=0.35). The difference in State-Trait Anxiety Inventory score was at p=0.04 at baseline but p=0.45 and p=0.83 after 7 and 55 days of treatment respectively. Therefore, psychological variables were not significantly different at baseline when correcting for multiple comparisons (at p=0.05/9) and the one passing the uncorrected threshold p=0.05 was not significantly different during the testing phase.

## Drug and testing schedule

Two centers participated in the study (Oxford and Manchester, UK). Data were collected between January 2012 and July 2013. Participants were randomly assigned to one of the two parallel treatment groups following a double-blind procedure: placebo or escitalopram (10 mg during week #1 and #9; 20 mg from week #2 to #8). The randomisation list was constructed in blind, by the Institut de Recherches Internationales Servier, with stratification by gender and center (Oxford, Manchester). Treatments were conditioned by Les Laboratoires Servier Industrie so as to be visually indistinguishable and shipped in numbered containers to the investigators. The randomisation list was not made available to the investigators until the final data set had been checked for quality and locked by transfer to a Contract Research Organization (Biotrial). Participants took a daily oral capsule around 8 P.M. for 9 weeks and performed the Effort Allocation Task three times at distinct latencies (*Figure 1B*): initial (2–4 days after treatment onset), intermediate (12–17 days) and late (52–60 days for all subjects but 3, who were tested between the 33rd and 44th days before withdrawing from the study). The task was not performed on week #9. The clinical trial included other tests, not presented here, to assess emotional processing, sexual acceptability and learning abilities. Safety evaluations (adverse events collection, blood pressure/heart rate and laboratory test) were performed along with the study.

## Experimental set-up

The Effort Allocation Task is schematized in *Figure 1A* and detailed in a previous publication (*Meyniel et al., 2013*). The only change was the adoption of the local currency (British pounds). On each trial, participants had 30 s that could be spent either resting or squeezing a handgrip. They were instructed that the payoff would be proportional to both the monetary incentive and the time spent above a force target corresponding to effort difficulty. The task lasted approximately one hour and was split into 8 blocks. The factor levels (monetary incentives: 1, 2 or 5p and effort difficulty: 70, 80, 90% of the maximal force) were manipulated independently and crossed, resulting in 9 conditions, each corresponding to a trial, presented in a randomized order in each block. Left and right hands were used alternatively over blocks. Participants were encouraged to maximize their payoff at each trial, and told that the cumulated payoff would be added to their financial compensation for participating in the study. Unbeknown to them, this payoff was rounded up to a fixed amount after the last visit such that all participants eventually received the same total. The task difficulty was adjusted to the subject's maximal force, which was measured at each visit before the test. The procedure for measuring the maximal force (sustained handgrip squeezing) is detailed in (*Meyniel et al., 2013*) and follows published guidelines (*Gandevia, 2001*). The maximal force was not affected by treatments ($F_{3, 113.7}$=0.93, p=0.42), visits ($F_{2, 180}$=1.34, p=0.26) or by the interaction of these two factors ($F_{6, 180}$=0.27, p=0.95).

## Behavioral analysis

The cumulated effort duration determined the payoff obtained at each trial. Yet there are many ways of chunking this cumulated duration depending on the duration of each effort and rest epochs. Effort and rest epochs were determined based on the force time series (*Figure 1A* provides an example) using an off-line algorithm (*Meyniel et al., 2014*). The offline detection algorithm was based on both the force signal normalized by the calibration maximal force, and its temporal derivative. Samples with positive derivative, exceeding one standard deviation of the derivative time series, and force level higher than 0.5 (half the maximum) were tentatively marked as effort onsets. Effort offsets were defined similarly for negative derivative values and force levels below 0.5. When multiple offsets were detected between two onsets, all but the last one were discarded. If multiple onsets were detected before an offset, the one with minimum force was kept. An offset was marked at the trial end if the effort was still sustained at that moment. Elapsed time between effort onsets and offsets determined effort and rest durations. The first rest duration was the elapsed time between trial onset and the first effort onset.

We performed a model-free analysis of the behavior, for each subject and visit, with multiple linear regressions done separately for effort and rest durations. The linear models included the factors of interest (incentive and difficulty) and temporal factors (block number; trial number within a block, effort or rest epoch number within a trial). Regressors (excepted the constant) were z-scored so that regression coefficients (beta estimates) correspond to standardized effect sizes. These beta values were then compared between treatments using ANOVA (see statistical analysis below).

## Computational analysis

### The cost-evidence accumulation framework

To account for the decision to stop and resume effort exertion in our task, we previously developed a computational model that was supported by both behavioral and neuroimaging data (*Meyniel et al., 2013*; *Meyniel et al., 2014*). The central assumption of our model is that the allocation of effort over time is underpinned by the variation of a single computational variable. This decision variable waxes during effort until reaching an upper bound, which triggers effort cessation, and wanes during rest until reaching a lower bound, which triggers effort resumption. The model thus has three latent parameters: the accumulation ($S_e$) and dissipation ($S_r$) slopes of cost evidence and the amplitude of these variations ($A$; the distance between bounds). The ratios of amplitude and slopes determine the effort and rest durations (see *Figure 2A*).

The computational analysis, detailed below, aimed at (1) identifying how the model latent parameters are modulated by the task factors (incentive and difficulty levels), (2) testing whether these modulations differ between treatment groups and (3) assessing the specificity and sensitivity of our fitting procedure in detecting potential differences between treatment groups.

### Model fit and selection

In this section, we provide details about model fit and model selection, although we followed the exact same methods as reported in our previous publication (*Meyniel et al., 2013*).

In principle, each model latent parameter (accumulation slope $S_e$, dissipation slope $S_r$ and amplitude of variations $A$) could be modulated by each task factor, which we formalized as linear effects:

$$Te = \frac{A}{Se}; Tr = \frac{A}{Sr};$$
$$\begin{cases} A & = A_m + A_i I + A_d D \\ Se & = Se_m + Se_i I + Se_d D \\ Sr & = Sr_m + Sr_i I + Sr_d D \end{cases} \tag{1}$$

where $T_e$ and $T_r$ are the mean durations of effort and rest epochs (fitted across experimental conditions); $I$ and $D$ are the z-scored incentive and difficulty levels; $A_i$, $A_d$, $Se_i$, $Se_d$, $Sr_i$ and $Sr_d$ capture potential modulations by task factors. The goal of model selection was to identify whether allowing fewer modulations, e.g. only $A_i$, $Se_d$ and $Sr_i$, could still provide a reasonable goodness-of-fit. In total, there are $2^6 = 64$ possible combinations of the six modulations by task factors. However, we noted in (*Meyniel et al., 2013*) that only 20 combinations can a priori reproduce the behavioral effects observed in the effort allocation task (and potentially other effects): increased $T_e$ with incentive,

decreased $T_r$ with incentive, and decreased $T_e$ with difficulty. These 20 models correspond to four possible combinations of the incentive effect: $\{A_i, Se_i, Sr_i\}$ or $\{A_i, Se_i\}$ or $\{A_i, Sr_i\}$ or $\{Se_i, Sr_i\}$ crossed with five possible combinations of the difficulty effect: $\{A_d, Se_d, Sr_d\}$ or $\{A_d, Se_d\}$ or $\{A_d, Sr_d\}$ or $\{Se_d, Sr_d\}$ or $\{Se_d\}$. Besides these modulation terms, all models also included the mean accumulation and dissipation slopes, $Sr_m$ and $Se_m$, as free parameters. Since effort and rest durations are determined by the ratios of amplitude and slopes in *Equation 1*, the mean amplitude $A_m$ was not considered as a free parameter; it was fixed to one, so that each subject could be characterized by a unique set of best-fitting parameter values instead of an infinity of proportional solutions.

We used Bayesian model selection to identify a solution with the best tradeoff between goodness-of-fit and simplicity (number of modulation terms in the model). The 20 models were fitted for each subject and visit separately using a variational Bayesian procedure described in (*Daunizeau et al., 2014*). We used non-informative priors for the fitted parameter values. This variational procedure provides, for each visit and model, the posterior mean values of the latent parameters and the model evidence. The latter is a key summary statistic on the basis of which the best model is identified (*Stephan et al., 2009*; *Daunizeau et al., 2014*). In order to obtain a single summary statistic per subject, we took the joint (i.e., product of) model evidence over visits. The algorithm used for group-level, random-effect Bayesian Model Selection is described in (*Stephan et al., 2009*) and implemented in the Matlab toolbox SPM8 (Wellcome Department of Imaging Neuroscience, London, UK; function spm_BMS.m).

## Best computational model identified

In both treatment groups, Bayesian model selection (see *Figure 2A*) identified the following model as providing the best trade-off between simplicity and goodness-of-fit of the data:

$$Te = \frac{A}{Se}; Tr = \frac{A}{Sr};$$
$$\begin{cases} A & = A_m + A_i I \\ Se & = Se_m + Se_d D \\ Sr & = Sr_m + Sr_i I \end{cases} \tag{2}$$

This is the same best model as in our previous publication (*Meyniel et al., 2013*), although it is fitted on different subjects. Fitted parameter values were on average positive such that higher incentives expanded the bounds of cost-evidence variations and quickened its dissipation during rest and higher difficulty levels increased the accumulation rate of cost-evidence during effort (see *Figure 2B*).

This computational model can be interpreted as follows, for a more complete discussion see (*Meyniel et al., 2013, 2014*). The accumulation of cost evidence may reflect the build-up of physiological fatigue: the instantaneous effort cost increases as it becomes gradually more difficult to sustain a given force level. As expected from a physiological perspective, fatigue builds up more quickly when higher force levels are exerted, which is captured by $Se_d$. An upper bound on the accumulation process ensures that instantaneous cost remains within a certain range that is adjusted to the reward rate. Indeed, the bound is pushed back for higher incentives (which is captured by $A_i$), as if an extra cost is allowed when a higher benefit makes it worthy. The bounded accumulation mechanism therefore seems to control the balance between cost and benefit, which could be implemented as an opponency between brain systems (*Daw et al., 2002*; *Balasubramani et al., 2015*).

The dissipation of cost evidence may reflect that it takes time to recover full exercising capacity after a strenuous effort. Interestingly, effort was not resumed immediately after cessation, but only after a substantial decrease in cost evidence. Waiting until a lower bound is reached may ensure that effort is not systematically produced at maximal instantaneous cost. Rest may therefore contribute to optimizing the cost/benefit balance on the long run. Importantly, an exceedingly long rest would also be suboptimal since it would reduce the payoff. Indeed, only efforts are rewarded in the task: there is therefore an opportunity cost to rest (*Niv et al., 2007*). This opportunity cost seems to be taken into account since the cost-evidence dissipation is speeded up for higher incentives (which is captured by $Se_i$), so that effort can be resumed more quickly.

## Specificity and sensitivity of the fitting procedure

The fitted parameter values were then compared between treatment groups (see below). We used simulations to assess whether our fitting procedure can detect specific changes in the model parameters between groups. For each subject in the placebo group, we fitted the model and took the mean value over visits of a given parameter (henceforth the 'target'). To simulate a potential effect of the SSRI treatment on this target parameter, we increased this value by a 20%. Then we computed mean effort and rest durations in each experimental condition using *Equation 2* and we corrupted these means with noise (we added the residuals of fitted data taken from another random subject in the placebo group). Last, we fitted again the model by adjusting a second set of parameters onto these simulated mean effort and rest durations. To assess statistical significance, we repeated this procedure 100 times for each subject and each target parameter. We compared the initial parameter values (fitted on actual data) and the new ones (fitted on simulated data) for each simulation with paired t-test across subjects and the threshold $p < 0.05/5$ to correct for the 5 parameters tested. For 'target' parameters, a significant difference indicates that the fitting procedure correctly recovers the simulated increase (a 'hit'); for 'non-target' parameters, a significant difference indicates a lack of specificity (a 'false alarm'). We report the simulated changes identified by the fitting procedure in *Figure 2C*, as the median parameter change (across subjects), averaged over simulations.

## Specificity of the treatment effect

We also used Bayesian model comparison to test the specificity of treatment effect on computational parameters. We performed this comparison at the group level since our design is between-subject and different subjects may have different parameter values irrespective of the treatment type. To this end, we concatenated across subjects the mean effort and rest durations per condition, separately for each group. The parameters of the accumulation model (*Equation 2*) were fitted onto the placebo group data. These fitted values served as reference: $A_i^{REF}$, $Se_m^{REF}$, $Se_d^{REF}$, $Sr_m^{REF}$, $Sr_i^{REF}$. We then modeled the potential treatment effect in the escitalopram group as a modulation of the reference parameters: $A_i = A_i^{REF}*(1+\delta_{Ai})$, $Se_m = Se_m^{REF}*(1+\delta_{Sem})$, and so on for $Se_d$, $Sr_m$ or $Sr_i$. For each parameter, the sign and value of $\delta$ captures the direction and strength of the treatment effect with respect to the placebo group. We designed single-modulation variants of the accumulation model (*Equation 2*) by forcing $\delta$ to zero for all but one parameter, for which $\delta$ remained free. In other words, a modulation of only one single parameter was permitted. We also designed double-modulation models in which two modulations are permitted and modeled with distinct $\delta$s. All models were fitted with the variational Bayesian procedure by (*Daunizeau et al., 2014*). Values of log model evidence are reported in *Table 3* for model comparison.

## Statistical analysis

We characterized the effect of treatments on (1) the monetary payoff, (2) the fitted parameters of the best computational model (*Equation 2*), (3) the regression coefficients of the model-free analysis. We provide source data files for each of these three sets of variables. We used a repeated-measure ANOVA with subjects as a random factor, treatment group as a between-subject factor and time since treatment onset (initial, intermediate, late) as a within-subject factor. Interaction between the within-subject factors was included in all ANOVAs. We followed up the ANOVA results with two-sample two-sided t-tests at a given time since treatment onset. Significance levels corrected for comparing multiple variables are included in *Table 2*.

Given our large sample size (N=58 included subjects), these classical parametric statistics reliably quantify the likelihood that the observed differences between treatment groups reported in our study may be due to chance in sampling the population. Indeed, we confirmed these significance levels with non-parametric permutation tests, using 10,000 permutations of treatment labels between subjects to estimate the probability that an equal or more extreme statistic (F or T depending on the test) could occur by chance.

## Additional information

### Competing interests
FM: Supported by The French Ministère de la Recherche and the European Union Seventh Framework Programme (FP7/2007-2013) under grant agreement no. 604102 (Human Brain Project). GMG: Holds shares in P1vital Ltd and has served as consultant, advisor or CME speaker in the last 12 months for AstraZeneca, Cephalon/Teva, Convergence, Eli Lilly, GSK, Lundbeck, Medscape, Otsuka, Servier, Sunovion, Takeda. JFWD: Currently advises or carries out research funded by Autifony, Sunovion, Lundbeck, AstraZeneca and Servier; all payment is to the University of Manchester; he has share options in P1vital Ltd. RG: Received compensation as a member of the scientific advisory board of Janssen, Lundbeck, Roche, Takeda; he has served as consultant and/or speaker for Astra Zeneca, Pierre Fabre, Lilly, Otsuka, SANOFI, Servier and received compensation, and he has received research support from Servier. The other authors declare that no competing interests exist.

### Funding

| Funder | Grant reference number | Author |
|---|---|---|
| Institut de Recherche Internationales Servier | Study No. CL1-20098-81 | Florent Meyniel<br>Guy M Goodwin<br>JF William Deakin<br>Corinna Klinge<br>Christine MacFadyen<br>Holly Milligan<br>Emma Mullings<br>Mathias Pessiglione<br>Raphaël Gaillard |
| French Ministère de la Recherche | PhD Fellowship | Florent Meyniel |
| Seventh Framework Programme | Postdoc fellowship (no. 604102) | Florent Meyniel |

The funders had no role in data collection, interpretation and publication.

### Author contributions
FM, Conception and design, Analysis and interpretation of data, Drafting or revising the article; GMG, JFWD, Conception and design, Drafting or revising the article; CK, Collected all data together with CMF from the Oxford center, Acquisition of data; CM, Collected all data together with CK from the Oxford center, Acquisition of data; HM, EM, Collected all data together with EM from the Manchester center, Acquisition of data; MP, Interpretation of the result, Conception and design, Drafting or revising the article; RG, Interpretation of the results, Conception and design, Drafting or revising the article

### Author ORCIDs
Florent Meyniel, http://orcid.org/0000-0002-6992-678X
JF William Deakin, http://orcid.org/0000-0002-2750-962X

### Ethics
Clinical trial registration ISRCTN75872983, URL: http://www.isrctn.com
Human subjects: The study was approved by the Ethics Committee of Berkshire, UK on 28th Sept. 2011 (protocol CL1-20098-81) and was registered as a clinical trial (ISRCTN75872983). Healthy participants gave their written informed consent prior to participation.

## Additional files

### Supplementary files
• Reporting standards 1. Consort 2010 checklist.

• Reporting standards 2. Consort 2010 flow diagram.

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
