## [Decision Letter]

Thank you for submitting your article "A specific role for serotonin in overcoming effort cost" for consideration by *eLife*. Your article has been reviewed by three peer reviewers, and the evaluation has been overseen by Joshua Gold as the Reviewing Editor and Sabine Kastner as the Senior Editor. The following individuals involved in review of your submission have agreed to reveal their identity: Read Montague (Reviewer #1); Roshan Cools (Reviewer #3).

The reviewers have discussed the reviews with one another and the Reviewing Editor has drafted this decision to help you prepare a revised submission.

Summary:

This study used a physical effort/incentive-based task with two groups of subjects – one taking escitalopram, the other placebo – to test whether SSRIs, and putatively serotonin, have an effect on the amount of effort human subjects will expend to get a reward. Several of the authors had previously developed a rise-to-bound computational model to account for behavior on this task in terms of a "cost evidence" variable that rises during effort and falls during rest. Here they used the model to show that overall better performance of the SSRI group (more reward) is associated with reduced effort costs and not the weight in monetary incentives. Thus, 5HT's role in action costs might be more general than hitherto thought, extending from the regulation of punishment and delay to that of effort costs.

The reviewers all agreed that this is an excellent paper: interesting, well done, and well written. The role of serotonin in behavioral control and decision-making is complex, and this paper makes a solid contribution to shrinking some of this complexity by sharpening our understanding of it in relation to effort. The ubiquity of SSRI use – in depression, Parkinson's Disease, and other disorders – and the lack of model-based understanding of what these drugs do in terms of information processing makes the contribution important. It is also commendable from a clinical perspective that chronic rather than just acute effects of the drug were assessed.

Essential revisions:

1) Several reviewers thought that the model needed to be explained better, and the modeling results analyzed more thoroughly.

A) Specifically, the text only briefly refers to specific parameters ("amplitude, accumulation and dissipation slopes of the cost-evidence decision variable") that a reader without thorough familiarity of the earlier papers will have a hard time following. It would be useful to describe the model in the text. Likewise, the model parameters are listed ("A_i_, Se_m_, etc.) without being explained. For more context, it might also be useful to relate their model more directly to the RL model in Daw et al. 2002 referenced in the paper, and also the model in Balasubramani et al. 2015, Frontiers of Computational Neuroscience.

B) More substantively, there were questions about the specificity of the effect from the modeling analyses. It is stated that interactions between treatments and pairs of computational variables (Se_m_ versus each other parameter) were always significant. The implication of this result depends on the sign of the parameters. The cumulative payoff effect might represent, for example, enhanced amplitude modulation (A_i_) and/or reduced cost-evidence accumulation (Se_m_). How were computational parameters coded when entered into the ANOVA? To assess specificity, one might want to reverse the coding for A and Sr. More generally, to what extent were the parameters independent of each other in the model? Also, might it be possible to define a different model space that would allow the use of Bayesian stats, and model comparison to estimate the drug effects on the various parameters? The distinct advantage of such an approach would be that it allows the capturing of drug effects despite the presence of individual differences of no interest.

C) Is any background 'reference' neuropsychological data available to confirm that there were no group differences of no interest? If so, please report.

2) There were questions about the interpretation of the effects in terms of the role of time in task performance. The main effects coming out of the modeling analysis appear to be related to effort duration. But longer effort durations are also longer overall durations, which brings up the literature on 5-HT and delay costs (some of the delay discounting papers with 5-HT manipulations are actually cited in the manuscript). The authors appear to favor the view that different types of costs can be reduced to one process that is related overall to multiple types of costs. There is evidence both for and against this, and the authors are of course free to interpret their results in this way. In fact, this discussion contributes to the field in an interesting way. However, how much of the effect they are seeing is related to time per se? The authors should discuss this.

3) The Introduction might benefit from mentioning the implication of 5HT not just in apathy, but also in the other end of the motivation continuum, that is impulsivity. And in the Discussion, how do the findings relate to observations that 5HT is key for impulse control? More generally, the paper might come full circle, by linking, in the Discussion section, the obtained results to these various clinical hypotheses (involvement in apathy, impulsivity).

[Editors' note: further revisions were requested prior to acceptance, as described below.]

Thank you for resubmitting your work entitled "A specific role for serotonin in overcoming effort cost" for further consideration at *eLife*. Your revised article has been favorably evaluated by Sabine Kastner (Senior editor), a Reviewing editor (Joshua Gold), and one new reviewer, who offered input from a statistical perspective. The manuscript has been improved but there are some remaining issues that need to be addressed before acceptance, as outlined by the reviewer:

*Reviewer #4:*

I have reviewed this paper from a statistical perspective as this is a registered trial.

This manuscript presents the analysis of a single randomised placebo controlled and double blinded study. It is not a trial of an intervention but rather a controlled study to understand the mechanism of action that the specific drug takes. The trial has been registered with a target sample size of 128 and it has a single hypothesis statement that aims to 'assess the effects of' agolmelatine and escitalopram on emotional binding, emotional processing and motivation'. Each participant is measured over several tasks during the treatment period of 9 weeks.

I realise that it is not in the *eLife* style to report numerical estimates in the Abstract but as this report analyses a single controlled experiment I think they should put the strength of the evidence in the Abstract to back up their statement that 'they show that serotonin also regulates other types of action costs such as effort'. As there is no replication they should report this in a more guarded manner.

The authors have completed the Consort statement but they need to provide more complete information within the manuscript so they can say yes to more of the checklist.

The trial registration lists a compound aim/hypothesis and also lists three primary outcomes listed which I think have all been used in this analysis but the main question for the manuscript is 'does serotonin regulate the weight of effort cost as opposed to the weight of expected benefit?'

I would like to see a clearer link between the main research question and the forms of analysis. A sample size of 58 is not that large even with repeated measurement. Was there really any power to test for interactions? The Consort checklist says 'no' to sample size but surely the size was justified originally? I feel a bit concerned that sometimes Bonferonni correction is used to conclude there were no differences when this just reduces power.

---

## [Author Response]

*Essential revisions:*

*1) Several reviewers thought that the model needed to be explained better, and the modeling results analyzed more thoroughly.*

*A) Specifically, the text only briefly refers to specific parameters ("amplitude, accumulation and dissipation slopes of the cost-evidence decision variable") that a reader without thorough familiarity of the earlier papers will have a hard time following. It would be useful to describe the model in the text. Likewise, the model parameters are listed ("A_i_, Se_m_, etc.) without being explained. For more context, it might also be useful to relate their model more directly to the RL model in Daw et al. 2002 referenced in the paper, and also the model in Balasubramani et al. 2015, Frontiers of Computational Neuroscience.*

We agree that the model should have been presented with more details in the main text, instead of being confined to the Methods section. We have revised the model description in the Results section, which now appears as follows:

“The better performance observed in the escitalopram group could be underpinned by different mechanisms: alleviation of effort costs or inflation of incentive values, or both. […] To capture all the effects, the cost-evidence accumulation model therefore necessitates five free parameters: the mean slope of cost-evidence accumulation (Se_m_) and its steepening for higher difficulty levels (Se_d_); the mean slope of cost-evidence dissipation during rest (Sr_m_) and its steepening for higher incentives (Sr_i_); and the expansion of the cost-evidence amplitude as higher incentives push the bounds back (A_i_).”

The referees also ask for a more detailed contextualization of our model, in particular with respect to the reinforcement learning framework (RL).

We would like to start by pointing important conceptual distinctions with respect to RL. First, RL deals with mapping stimuli to responses and outcomes across time. The parallel with our task is limited since here, external stimuli are mostly restricted to cues that have an explicit meaning (coins indicating the reward rate) and outcomes are deterministically and immediately coupled to actions: the cumulative payoff is displayed continuously as a function of the current force and incentive level. Second, the dynamics in RL reflects the learning of stimulus-response-outcome mapping across trials, whereas in our task, the dynamics reflects the allocation of effort within and between trials. Third, RL usually comes with a discretization of events into trials, with transient prediction errors whereas in our task, actions and feedback are continuous in time. Last, in RL, “cost” refers to negative outcomes that thereby vary along the same axis as “benefit” with a negative valence. In contrast, the effort cost is instrumental to get reward in our task. In total, the processes that we intended to model here do not involve much RL, as the mapping is clearly specified in the instructions. These processes are meant to solve a choice problem: knowing all the contingencies, how should I allocate my effort over time?

The referees also point to Balasubramani et al. 2015 who provide a detailed model for the implementation of RL in the basal ganglia, adding the notion of risk to standard RL. This level of detail is far beyond the sort of description that is needed to capture the behavior in our task. We acknowledge that the uncertainty about one's ability to sustain effort in the future could be seen as similar to the notion of risk defined as the variance in potential outcomes. However, such a link is extremely speculative for the moment.

Therefore, we prefer to avoid making a direct comparison to RL models. However, we intended to contextualize our model with respect to relevant notions that emerged in the field of RL, such as the notions of opponency, optimization, time scale and opportunity cost. We added the following paragraph in the Methods (with reference to Balasubramani et al., 2015):

“This computational model can be interpreted as follows, for a more complete discussion see (Meyniel 2013, 2014). The accumulation of cost evidence may reflect the build-up of physiological fatigue: the instantaneous effort cost increases as it becomes gradually more difficult to sustain a given force level. […] This opportunity cost seems to be taken into account since the cost-evidence dissipation is speeded up for higher incentives (which is captured by Se_i_), so that effort can be resumed more quickly.”

*B) More substantively, there were questions about the specificity of the effect from the modeling analyses. It is stated that interactions between treatments and pairs of computational variables (Se_m_ versus each other parameter) were always significant. The implication of this result depends on the sign of the parameters. The cumulative payoff effect might represent, for example, enhanced amplitude modulation (A_i_) and/or reduced cost-evidence accumulation (Se_m_). How were computational parameters coded when entered into the ANOVA? To assess specificity, one might want to reverse the coding for A and Sr. More generally, to what extent were the parameters independent of each other in the model? Also, might it be possible to define a different model space that would allow the use of Bayesian stats, and model comparison to estimate the drug effects on the various parameters? The distinct advantage of such an approach would be that it allows the capturing of drug effects despite the presence of individual differences of no interest.*

Parameters were entered in their native format in the ANOVA reported in the submitted manuscript. We agree that the direction of the effect matters. After flipping the sign of Se_m_, the interaction between variable types (Se_m_ vs. other variables) and groups (Placebo vs. Escitalopram) becomes:

Se_m_:A_i_ p=0.295

Se_m_:Se_d_ p<0.001

Se_m_:Sr_m_ p=0.68

Se_m_:Sr_i_ p=0.27

However, flipping signs is rather post-hoc and motivated by seeing the actual results. Since this approach is questionable, we adopted, instead, a Bayesian model selection (BMS) approach. Indeed, as suggested by the reviewers, BMS may be more suited to address the issue of specificity, irrespective of the direction of changes induced by the treatment. We therefore replaced the paragraph with the ANOVA result by a model comparison.

In short, this comparison shows that there is a specific modulation of the latent parameters of cost-evidence accumulation by the SSRI treatment. Among single-modulation models, a modulation of Se_m_ outperforms modulation of any other parameter. Among double-modulation models, modulations of Se_m_ and Se_d_ outperform any other combination. These best models are also much better than a null model (without any modulation).

We replaced the paragraph with the ANOVA results by the following paragraph:

“To further test the specificity of SSRI effect on cost-evidence accumulation, we performed another Bayesian model selection that contrasted the two groups of subjects. [...] A version with modulations of both Se_m_ and Se_d_ was slightly more likely than a version with only a modulation of Se_m_ (Δ=3) and much more likely than any other combination (Δ>5.9).”

We added the Table 3 to the revised manuscript.

We also added the following paragraph in the Methods section:

“Specificity of the treatment effect

We also used Bayesian model comparison to test the specificity of treatment effect on computational parameters. […] All models were fitted with the variational Bayesian procedure by Daunizeau et al. 2014. Values of log model evidence are reported in Table 3 for model comparison.”

Regarding whether model parameters are independent from one another, the equation of the model suggests that they are not. For instance, the sensitivity of effort duration to incentives is controlled by the parameter A_i_ and scaled by Se_m_. However, the model is identifiable: for a given behavior, there is only one best-fitting set of parameter values. The critical issue for our pharmacological study is whether our model and fitting procedure can detect specific changes induced by the treatment. We quantified the specificity and sensitivity of our method with simulations (Figure 2). We unpack this aspect further and the related paragraph now appears as:

“We checked the sensitivity and specificity of our model fitting procedure in detecting a treatment effect through simulations. A 20% change in any given parameter was reliably detected and the difference recovered by model fitting was significant in a 96% of simulations at least for A_i_, Se_m_, Sr_m_ and Sr_i_ and in a 77% of simulations for Se_d_. Importantly, a change in one single parameter was recovered without propagating to other parameters and the false alarm rate was below 5% for all parameters (Figure 2).”

*C) Is any background 'reference' neuropsychological data available to confirm that there were no group differences of no interest? If so, please report.*

Indeed, participants were scored on psychometric scales. We included the following paragraph in the Methods:

“We also used psychological tests to assess differences between groups of subjects at baseline, before treatment. […] Therefore, psychological variables were not significantly different at baseline when correcting for multiple comparisons (at p=0.05/9) and the one passing the uncorrected threshold p=0.05 was not significantly different during the testing phase.”

*2) There were questions about the interpretation of the effects in terms of the role of time in task performance. The main effects coming out of the modeling analysis appear to be related to effort duration. But longer effort durations are also longer overall durations, which brings up the literature on 5-HT and delay costs (some of the delay discounting papers with 5-HT manipulations are actually cited in the manuscript). The authors appear to favor the view that different types of costs can be reduced to one process that is related overall to multiple types of costs. There is evidence both for and against this, and the authors are of course free to interpret their results in this way. In fact, this discussion contributes to the field in an interesting way. However, how much of the effect they are seeing is related to time per se? The authors should discuss this.*

We agree that time is an essential dimension in our task. However, the cost is about effort, not delay. This is because prolonging effort duration does not postpone reward delivery. Although the effect of SSRI accords well with the general idea that serotonin helps with overcoming cost, it cannot be explained by a change in temporal discounting. We intended to clarify our interpretation by adding the following paragraph to the Discussion:

“It is important to note that the SSRI effects obtained here relate to effort cost and not to time discounting. Although time is central in our task, it departs from delay and waiting paradigms in several respects. [...]Therefore, the SSRI effect on effort duration (how long subjects sustain an effort) cannot be explained by a change in temporal discounting (how long subjects wait for a reward).”

*3) The Introduction might benefit from mentioning the implication of 5HT not just in apathy, but also in the other end of the motivation continuum, that is impulsivity. And in the Discussion, how do the findings relate to observations that 5HT is key for impulse control? More generally, the paper might come full circle, by linking, in the Discussion section, the obtained results to these various clinical hypotheses (involvement in apathy, impulsivity).*

We agree that impulsivity is a core concept in the literature on serotonin function. The Introduction implicitly referred to impulse control when quoting effects on behavioral inhibition and patience for delayed reward. We have now included this keyword in the text.

In the Discussion, we also integrated a reference to impulse control in the paragraph mentioning the effects of serotonin in waiting tasks. We also included a new paragraph to link our hypothesis and results to apathy and impulsivity:

“A general role for serotonin in cost processing also seems compatible with the effects on apathy and impulsivity reported in healthy and pathological conditions (Cools et al. 2005; Papakostas et al. 2006; Schweighofer et al. 2008; Crockett et al. 2009; Weber et al. 2009; Warden et al. 2012; Guitart-Masip et al. 2014; Miyazaki et al. 2014; Worbe et al. 2014). […] As SSRIs might also reduce the perceived cost of delaying action, they would improve response control and reduce impulsivity (Cools et al. 2005, 2011; Crockett et al. 2009; Dayan 2012; Bari and Robbins 2013; Miyazaki et al. 2014; Fonseca et al. 2015).”

[Editors' note: further revisions were requested prior to acceptance, as described below.]

*The manuscript has been improved but there are some remaining issues that need to be addressed before acceptance, as outlined by the reviewer:*

*Reviewer #4:*

*I have reviewed this paper from a statistical perspective as this is a registered trial.*

*This manuscript presents the analysis of a single randomised placebo controlled and double blinded study. It is not a trial of an intervention but rather a controlled study to understand the mechanism of action that the specific drug takes. The trial has been registered with a target sample size of 128 and it has a single hypothesis statement that aims to 'assess the effects of' agolmelatine and escitalopram on emotional binding, emotional processing and motivation'. Each participant is measured over several tasks during the treatment period of 9 weeks.*

*I realise that it is not in the eLife style to report numerical estimates in the Abstract but as this report analyses a single controlled experiment I think they should put the strength of the evidence in the Abstract to back up their statement that 'they show that serotonin also regulates other types of action costs such as effort'. As there is no replication they should report this in a more guarded manner.*

We let the *eLife* editorial board decide whether this change is compatible with their style.

*The authors have completed the Consort statement but they need to provide more complete information within the manuscript so they can say yes to more of the checklist.*

We revised several aspects of the manuscript (see details below), which allowed us to check the following items of the Consort list: 7a, 8a, 9, 10, 14.

We also report the dates of the trial (item 14) in the section “Drug and Testing Schedule”: “Data were collected between January 2012 and July 2013.”

*The trial registration lists a compound aim/hypothesis and also lists three primary outcomes listed which I think have all been used in this analysis but the main question for the manuscript is 'does serotonin regulate the weight of effort cost as opposed to the weight of expected benefit?'*

*I would like to see a clearer link between the main research question and the forms of analysis. A sample size of 58 is not that large even with repeated measurement. Was there really any power to test for interactions? The Consort checklist says 'no' to sample size but surely the size was justified originally? I feel a bit concerned that sometimes Bonferonni correction is used to conclude there were no differences when this just reduces power.*

The trial registration lists three points. Point #1 and #2 are about tests on emotions performed by other research groups, not reported here. Point 3 lists goals for three “motivation” tasks, one being the Effort Allocation Task presented in this article, and the two others being probabilistic learning tasks that involved other researchers. Trial registration forms are highly stereotyped, and do not really fit the kind of exploratory trial that we participated in, with many tasks from various researchers. Therefore, our study addresses a much more specific question than the entire trial.

The sample size was proposed by the coordinators of the clinical trial, without formal calculation. This number seemed reasonable given typical studies in the field of cognitive neuroscience resorting to drugs in healthy subjects.

We added the following sentences in the Methods: “Given that the clinical trial was exploratory and also included tests from other research groups, the sample size was not selected specifically for our study; however, it appeared reasonable given typical studies in the field. Indeed, published between-subject comparisons of placebo and anti-depressant treatments were made on a lower sample size per group (e.g. N=14 in Harmer et al. 2014, N=20 in Chamberlain et al. 2006 and N=30 in Guitart-Masip et al. 2014), and a single visit per subject (while we have three)”.

We modified the Consort checklist, checking the corresponding item (7a) and referring to the relevant page of the manuscript.

Regarding the Bonferonni correction, we believe that one must correct for multiple comparisons to take into account the unwanted inflation of type I error. To this end, we used a Bonferonni correction, which is the most conservative one. The reviewer suggests that a Bonferonni correction may be used to conveniently discard marginally significant effects of little interest. However, note that it imposes an equally stringent test for the effects of interest. In any case, we treated all effects equally, by reporting non-corrected significance levels, and when there were multiple comparisons, by highlighting which significance levels passed the Bonferonni-corrected threshold. We believe it is both transparent and statistically correct.

We nevertheless agree with the reviewer that a lack of power is an issue for marginally or non-significant results. We added the following sentence in the Discussion section, regarding the interaction with time that we could not detect: “… our study may lack the statistical power to reveal such an effect”.